# Diminishing Returns Shape Constraints for Interpretability and Regularization

**Maya R. Gupta, Dara Bahri, Andrew Cotter, Kevin Canini**
Google AI
1600 Charleston Rd
Mountain View, CA 94043
{mayagupta,dbahri,acotter,canini}@google.com

## Abstract

We investigate machine learning models that can provide diminishing returns and accelerating returns guarantees to capture prior knowledge or policies about how outputs should depend on inputs. We show that one can build flexible, nonlinear, multi-dimensional models using lattice functions with any combination of concavity/convexity and monotonicity constraints on any subsets of features, and compare to new shape-constrained neural networks. We demonstrate on real-world examples that these shape constrained models can provide tuning-free regularization and improve model understandability.

## 1 Introduction

*Diminishing returns* are common in physical systems, human perception and psychology [1], and have been recognized in economics [2] and agriculture [3] for centuries. For example, a model that predicts how much a renter will like an apartment should predict a strong preference for 60 square meters of living space over 50 square meters, but a smaller preference for 100 square meters over 90 square meters, if everything else is the same. Similarly, a model that predicts the time it will take a customer to grocery shop should decrease in the number of cashiers, but each added cashier reduces average wait time by less. In both cases, we would like to be able to incorporate this prior knowledge by constraining the machine learned model's output to have a diminishing returns response to the size of the apartment or number of cashiers. Mathematically, we say a function has diminishing returns with respect to an input if the function is monotonically increasing and concave, or monotonically decreasing and convex, with respect to that input. *Accelerating returns* are also common in the real-world; for example, Adam Smith characterized labor specialization and economies of scale as causing accelerating returns [4]. Accelerating returns describes functions that are monotonically increasing and convex, or monotonically decreasing and concave with respect to an input.

We show how one can train flexible models that capture one's prior knowledge or preference that the model should exhibit diminishing or accelerating returns with respect to some inputs. Such *shape constraints* are effective regularization, reducing the chance that noisy training data or adversarial examples produce a model that does not behave as expected, and act as a machine learning poka-yoke (mistake-proofing) strategy [5] when a model is re-trained with fresh data. Unlike most regularizers, shape constraints do not require tuning the amount of regularization (beyond the binary decision of whether to apply a shape constraint), and are especially useful when there is domain shift between training and test distributions. Shape constraints have a clear semantic meaning, and thus improve interpretability because the user knows and understands at a high-level how each of the shape-constrained inputs affects

the output. We have found in practice that shape-constrained machine-learned models are much easier to debug and analyze.

We investigate diminishing returns shape constraints for two flexible, nonlinear function classes: neural networks and lattices. Real-world experiments illustrate how these shape constraints can be useful and effective in practice. However, consistent with the past literature on *monotonic* neural networks, we found it difficult to control the shape constraints on neural networks as flexibly as with the lattice models. Specifically, for our shape-constrained neural networks (SCNNs) we could not produce a monotonic response for an input without also constraining it to be convex/concave, and we must select either convex or concave constraints - that is, we cannot impose both for different features. In contrast, we show that one can shape-constrain lattice models for any mixture of monotonicity and concavity/convexity constraints on any subsets of the features, and achieve similar (and more stable) test metrics than with unconstrained deep neural networks on a breadth of real-world problems.

Another difference is joint vs *ceterus paribus* convexity. Our SCNNs impose *joint* convexity/concavity over all constrained features, whereas our lattice models' shape constraints are *ceterus paribus*: a constraint holds with respect to changes in a single feature if none of the other features change, but the convexity/concavity need not hold jointly over all the constrained features. For example, suppose $f(x)$ estimates the success of a party given two features: the number of guests and the number of bottles of wine. With the proposed lattice models, one can constrain $f(x)$ to have a ceterus paribus diminishing returns response in the number of guests (for any fixed number of bottles) or in the number of bottles of wine (for any fixed number of guests), but without forcing $f(x)$ to have a diminishing returns response along the diagonal direction defined as one bottle of wine per guest. In contrast, with the proposed SCNN, imposing diminishing returns on both wine and guests produces joint concavity, such that $f(x)$ will be concave and decreasing over any line in the two-dimensional feature space. We have found it easier to decide when ceterus paribus convexity is warranted for a problem. Joint convexity is a stronger constraint, and we have found it harder to reason about when it is warranted for real problems.

## 2   Related Work

We review the most related literature, categorized by function class. Note the term *partial* is used to mean that one can select which of the inputs is shape-constrained. Many of the function classes discussed below and in our proposals are multi-layer functions such that $f(x) = h(g(x))$. Recall from the chain rule that $f'' = h''(g')^2 + g''h'$. Thus if $h(x)$ and $g(x)$ are convex and increasing, then $f(x)$ is convex and increasing, and analogously for concave and increasing. Also, if $g(x)$ is convex and $h(x)$ is convex and increasing, that is sufficient for $f(x)$ to be convex.

**GAMs:** Generalized additive models (GAMs) [6] are a classic function class for imposing shape constraints [7]. Recall GAMs are a sum of component-wise 1-d nonlinear functions such that $f(x) = \sum_{d=1}^{D} f_d(x[d]) + b$, where $x \in \mathbb{R}^D$, each $f_d : \mathbb{R} \to \mathbb{R}$, and $b \in \mathbb{R}$ is a constant. Shape constraints are enforced by choosing appropriate parametric forms for each feature's function, $f_d$, that obey the desired constraints (e.g., for diminishing returns, use a positively-scaled log function). Also well-studied are nonparametric shape-constrained GAMs [7]; for example, the special case of $D = 1$ with monotonicity regularization is well-known as *isotonic regression* [8]. Recently, Chen et al. [9] gave an algorithm for fitting nonparametric GAM models with diminishing/accelerating returns constraints. Pya and Wood [10] also trained GAMs with first and second derivative shape constraints (see their paper for additional earlier work). These two methods for achieving diminishing returns performed similarly on a real dataset ($N = 915$ examples, $D = 4$ features) presented in Chen et al. [9], and both were notably better than unconstrained GAMs.

**Max-Affine (Max-Pooling):** Convex piecewise-linear models [11, 12] take advantage of the fact that any convex piecewise-linear function can be expressed as a max-affine function [13] (analogously, use min for concavity): $f(x) = \max_k \{A_k^T x + b_k\}$. Earlier, Sill [14] used max-affine functions followed by a min-pooling layer to form a three-layer *min-max network* to

learn monotonic functions, $f(x) = \min_j \max_k \{A_{jk}^T x + b_{jk}\}$, with the appropriate components of $A$ constrained to be positive.

**Neural Networks:** One can constrain a neural network to be monotonic by restricting the network weights to all be positive [15, 16, 17, 18, 19]. However, this strategy significantly reduces expressibility [20]. Specifically, if you constrain a neural net with *ReLU* activations to be increasing in $x$ by constraining its weights to be non-negative, then an annoying side-effect is that $f(x)$ will also be convex in $x$. Experimentally, monotonic neural nets have not performed as well as monotonic min-max networks or monotonic deep lattice networks [21].

Recently, Amos et al. [22] produced neural networks with partial convex shape constraints (but without fully enabling monotonicity shape constraints), focused on the goal of learning jointly convex functions that are easy to minimize. To achieve convexity, they add monotonicity constraints to weights in the later layers of a neural network using standard convex activation functions. For a single-layer and using max-pooling as the activation, this reduces to the same function class as convex piecewise-linear fitting [11, 12]. To make their architecture more flexible, they add an unrestricted linear embedding of the inputs into each of the later layers.

Dugas et al. [18] proposed an accelerating returns neural network that requires monotonicity for *all* inputs, and with partial convexity.

**Lattice Models:** Recent work in monotonic shape constraints has used multi-layer lattice models [21, 23, 24] (for an open-source implementation see `github.com/tensorflow/lattice`). Lattices are interpolated look-up tables where the look-up table parameters defining the function are learned with empirical risk minimization [25]; lattice models can also be expressed as multi-dimensional splines with fixed knots [26]. Lattice functions can be constrained to be partially monotonic by constraining adjacent parameters in the underlying look-up table to be monotonic for the shape-constrained inputs [24]. Ensembles of lattices [23] and deep lattice networks (DLN) [21] can similarly be constrained for monotoncity. Experimental results [21] on 4 real datasets showed monotonic DLNs with 3-4 layers performed similarly or better than min-max networks and simpler 2-layer ensembles of lattices, and notably better than monotonic neural networks.

## 3   Shape-Constrained Neural Network

We extend the partial *convex* neural network of Amos et al. [22] to enable partial *diminishing/accelerating returns* constraints. Without loss of generality, flip any decreasing features so that they are increasing before applying the following. Partition the feature vector $x$ into three subsets according to the desired shape constraints, $x_u$, $x_c$, and $x_s$, where we constrain the output $f(x; \theta)$ to be convex (concave) with respect to $x_c$, both convex (concave) and increasing with respect to $x_s$, and impose no constraints on $x_u$. Our $T$-layer SCNN is then $f(x; \theta) = z_T$, where each layer is defined by the following recurrence (illustrated in Figure 1):

$$u_{i+1} = h_i \left( W_i^{(1)} u_i + b_i^{(1)} \right) \quad \text{and} \quad u_0 = x_u$$

$$z_{i+1} = g_i \left( W_i^{(2)} \left( z_i \circ \left\lfloor W_i^{(3)} u_i + b_i^{(3)} \right\rfloor_+ \right) + W_i^{(4)} \left( x_s \circ \left\lfloor W_i^{(5)} u_i + b_i^{(5)} \right\rfloor_+ \right) \right.$$

$$\left. + W_i^{(6)} \left( x_c \circ \left( W_i^{(7)} u_i + b_i^{(7)} \right) \right) + W_i^{(8)} u_i + b_i^{(8)} \right) \quad \text{and} \quad z_0 = 0,$$

$$\text{where} \quad W_i^{(2)} \geq 0, W_i^{(4)} \geq 0. \tag{1}$$

where $\circ$ refers to the Hadamard product and $\lfloor x \rfloor_+ = \max(x, 0)$, $h_i$ can be any activation function but $g_i$ must be an increasing and convex (concave) activation to get convex (concave) shape constraints; like Amos et al. [22] we use $\text{ReLU}(x)$ for convexity; for concavity we recommend $-\text{ReLU}(-x)$. As is standard, for the last layer there is no activation function, i.e. $g_T$ is the identity function. We augment the architecture proposed in Amos et al. by adding new terms to support $x_s$ and new constraints on $g_i$ and $W_i^{(2)}$ to ensure diminishing/accelerating returns. This architecture works as described by the chain rule

and induction. Note that one cannot ask (1) to be convex with respect to some inputs and concave in others. Also, partial monotonicity can be imposed, but it comes with a side effect of convexity/concavity. Lastly, if more than one feature is constrained to be convex/concave, then (1) exhibits *joint convexity/concavity*: the function will be convex/concave over any line in the constrained feature space.

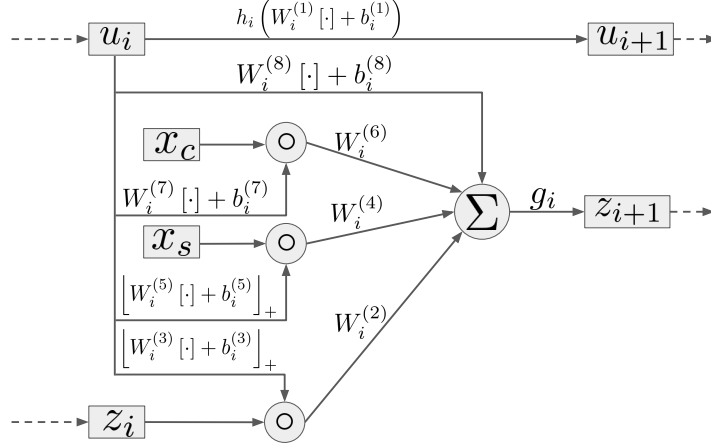

Figure 1: SCNN Architecture

# 4 Lattice Models with Convex/Concave Shape Constraints

We extend partial monotonic lattice ensemble models [23] [24] to also enable ceteris paribus partial convexity or concavity constraints, and thus also partial diminishing/accelerating returns constraints. The proposed lattice models can be individually constrained per-feature for any mix of these shape constraints. Let $x[d]$ be the $d$th component of the feature vector $x$, and in this section we require that each feature is bounded and has been pre-scaled per-feature such that $x[d] \in [0, 1]$ for all $d$.

## 4.1 Calibrated Linear Models with Convex/Concave Shape Constraints

We start with the simplest lattice model, called a *calibrated linear* model [24], which is a type of GAM:

$$f_{\beta,\alpha}(x) = \sum_{d=1}^{D} \alpha_d^T c_{\beta_d}(x[d]), \tag{2}$$

where the $D$ calibrated inputs $c_{\beta_d}(x[d])$ are linearly combined with coefficents $\alpha_d \in \mathbb{R}$, which can be individually constrained to be positive or negative if monotonicity is desired. Each calibrator function $c_{\beta_d} : [0, 1] \to [-1, 1]$ is a piece-wise linear function with $K - 1$ pieces, which we express as the linear interpolation of a one-dimensional look-up table with $K$ key-value pairs $(\nu_d[k], \beta_d[k])$ for $k = 1, \ldots, K$, where the keys $\{\nu_d[k]\}$ are fixed at the centers of the $K - 2$ uniform quantiles of the training data, and the choice of $K$ is a hyperparameter. See Fig. 3 for example calibrators.

To make $f(x)$ monotonic increasing with respect to the $d$th input, one must constrain the adjacent look-up table parameters of the $d$th calibrator $c_{\beta_d}$ to be increasing [24]:

$$\beta_d[k] \geq \beta_d[k-1] \text{ for } k = 2, 3, \ldots, K. \tag{3}$$

To make $f(x)$ concave with respect to the $d$ input, we must constrain the slopes in the piecewise linear function to be decreasing from the left, which requires that the differences in the calibrator's 1-d $K$-valued look-up table parameters be decreasing from the left:

$$\frac{\beta_d[k-1] - \beta_d[k-2]}{\nu_d[k-1] - \nu_d[k-2]} \geq \frac{\beta_d[k] - \beta_d[k-1]}{\nu_d[k] - \nu_d[k-1]} \text{ for } k = 3, 4, \ldots, K. \tag{4}$$

We fix the calibrator keys $\nu$ at the quantiles of the training data, so (4) is simply a linear inequality constraint on the three model parameters $\beta_d[k], \beta_d[k-1], \beta_d[k-2]$, for $k = K, K-1, \ldots, 3$.

For diminishing returns with respect to the $d$th input, one constrains the $d$th calibrator to be both increasing and concave by applying both (3) and (4) during training. Analogous constraints are needed to guarantee $f(x)$ convexity and accelerating returns. The model will guarantee the requested convex/concave response within the input bounds, but for input values outside the input's specified range, the model will clip the input value to the specified range, which makes the function implicitly flat outside the range, thus the accelerating returns guarantee only holds for inputs less than the input's specified maximum, and the diminishing returns only holds for inputs greater than the input's specified minimum.

Calibrated linear models are GAMs, with the special case of setting $K = N$ corresponds to a nonparametric model, though in practice we find the validated choice of $K$ is usually $5 - 50$, making calibrated linear models much more efficient to evaluate than non-parametric models. Thus, we expect shape-constrained calibrated linear models to perform similarly to the shape-constrained nonparametric GAMs of Chen et al. [9] and Pya and Wood [10].

## 4.2   Two Layer Lattice Models with Convex/Concave Shape Constraints

Next, we show that how to shape-constrain models with multi-dimensional lattices. While our proposal can be extended to deep lattice networks (see Appendix B), we focus on the two-layer lattice network formed by an ensemble of $L$ calibrated lattices [23]:

$$f_{\beta,\theta,W}(x) = \sum_{\ell=1}^{L} \theta_\ell^T \phi \left( W_\ell [c_{\beta_{\ell 1}}(x[1]) \ c_{\beta_{\ell 2}}(x[2]) \ \ldots \ c_{\beta_{\ell D}}(x[D])]^T \right), \qquad (5)$$

with definitions as follows. The $\ell$th base model calibrates the $d$th input using $c_{\beta_{\ell d}} : [0,1] \to [0,1]$ as defined in Sec. 4.1, except here we bound its output to $[0,1]$ so that the 2nd layer inputs lie in the unit hypercube. Next, each linear embedding $W_\ell \in \mathbb{R}^{S \times D}$ outputs $S$ values. The function $\phi : [0,1]^S \to [0,1]^{2^S}$ is a fixed kernel that transforms its $S$-dimensional input into the appropriate linear interpolation weights on the $2^S$ look-up table parameters for the $\ell$th lattice: $\theta_\ell \in \mathbb{R}^{2^S}$, for each of the $L$ lattices in the ensemble. The formula $\phi$ depends on which linear interpolation rule one uses [24]: (i) standard multilinear interpolation produces a multilinear polynomial on $z_\ell$ but is $O(2^S)$, (ii) the Lovász extension (aka *simplex interpolation*) produces a locally linear interpolation with $S!$ pieces and is $O(S \log S)$. We restrict attention to $2^S$ parameter look-up tables (see Appendix A for details on handling finer-grained look-up tables). We use the random tiny lattices (RTL) strategy to architect the ensemble of lattices [23], which means that before training we randomly select a fixed random subset of $S$ of the $D$ features for the $\ell$th lattice and fix the coefficients of $W_\ell$ to be $\{0,1\}$ to select the chosen $S \leq D$ features, where $S$ is a hyperparameter.

To produce convex/concave $f(x)$, by linearity it is sufficient to constrain each base model in the ensemble (5) to be convex/concave. If one interpolates the look-up table with the standard multilinear interpolation (by choosing the multilinear interpolation kernel $\phi$), then the fitted surface forms a multilinear polynomial over $[0,1]^S$, and is thus ceteris paribus linear in each feature (but nonlinear overall), and thus the lattice layer does not affect convexity/concavity (note this only holds because $W$ acts as a feature selector, (see Appendix B) for more on that). Thus, what is needed to guarantee that $f(x; \beta, \theta)$ is convex/concave with respect to the $d$th input is that each of the calibrators $\{c_{\beta_{\ell d}}(x[d])\}$ is convex/concave, and each of the $T$ look-up tables is monotonically increasing with respect to $x[d]$. For the monotonicity constraints, adjacent parameters in the look-up tables need to be constrained to be monotonically increasing, which is done by adding the appropriate linear inequality constraints on pairs of parameters (see [24]). What remains is to constrain the 1-d piecewise linear calibrators $\{c_{\beta_{\ell d}}(x[d])\}$ to be convex/concave, which requires constraining the calibrator values as per (4). If simplex interpolation is used for $\phi$ instead of multilinear interpolation, constraining for convexity/concavity requires additional constraints on the lattice parameters; see Appendix C for details.

### 4.3 Training the Constrained Optimization

The lattice models are trained using constrained empirical risk minimization, with the necessary constraints defined above to impose the selected shape constraints. Each of those constraints is a linear inequality constraint on two or three model parameters. For the calibrators, there are not very many shape constraints and we simply use projected SGD, where the Euclidean projection is performed by solving the resulting quadratic program, and we project onto all the calibrator shape constraints after each minibatch of 1000 stochastic gradients.

For the RTL models, to constrain the lattices for monotonicity, we must potentially satisfy a very large number of linear inequality constraints on pairs of adjacent lattice parameters, so solving the QP is less practical. Instead, we use the Light Touch algorithm [27] to stochastically sample the constraints, on top of Adagrad [28].

## 5 Experiments

We demonstrate the applicability and effectiveness of diminishing returns and accelerating returns regularization on five real-world regression problems (three of the five datasets are publicly available) with squared error loss (see also Appendix D for simulations). We compare: (i) a standard unconstrained DNN, (ii) the partial convex neural network [22], (iii) our shape-constrained neural network as per (1), (iv) calibrated linear models [24], (v) calibrated linear models with the proposed added convexity/concavity constraints, (vi) random tiny lattices (RTLs) [23] which in all cases used monotonic calibrators, (vii) RTLs with the proposed added convexity/concavity constraints.

Our TensorFlow code used for the SCNN is given in Appendix F. For the lattice models, we used proprietary C++ code to train, as described in Sec. 4.3. Similar results can be achieved by using the open-source Tensor Flow Lattice package (github.com/tensorflow/lattice), which already handles monotonicity constraints, and then adding an additive regularizer to penalize violations of (4).

For all model types and each set of constraints, the number of epochs and step sizes hyperparameters were optimized based on the validation set. All neural net models were run with TensorFlow using the Adam optimizer; see Sec. 4.3 for lattice model optimization. For all neural network models, the number of layers and number of units per layer were also validated from 1-9 and 3-1000 respectively. For all the lattice models, the number of keypoints in each of the calibrators was validated with one hyperparameter for all calibrators, from $K = 10, 20, 40, \dots$. For the RTLs, the number of lattices in the ensemble $L$, and the number of features per lattice $S$, were validated. Because the number of different validation options across model types was different, there was a risk that the best validated model was simply a particularly good random model. To control for that risk, after the hyperparameters were validated, each model type was freshly re-trained once with the validated hyperparameters, and the test error reported.

### 5.1 Car Sales

For this tiny 1-d problem with 109 training, 14 validation, and 32 test examples (`www.kaggle.com/hsinha53/car-sales/data`), we predict monthly car sales (in thousands) from the price (in thousands). Because it is a 1-d problem, the RTL model is the same as a calibrated linear model, and so was not separately run. Figure 2 shows the more constrained models are smoother and more interpretable, because a human can summarize the machine learned as, "Higher price cars decrease sales, but absolute price differences matter less the higher the price." Table 1 shows the Test MSE is slightly better for the calibrated linear model with the added convexity constraint. The convex SCNN was already decreasing in this case; the extra decreasing monotonicity constraint did not hurt.

Table 1: Experimental Results: Car Sales and Puzzle Sales

| Car Sales | | | Puzzles Sales | | |
|---|---|---|---|---|---|
| Model | Val. MSE | Test MSE | Model | Val. MSE | Test MSE |
| DNN | 2035 | 10931 | DNN | 2189 | 5652 |
| SCNN conv. | 2262 | 10613 | SCNN conc. | 2632 | 7931 |
| SCNN conv. decr. | 2442 | 10590 | SCNN conc. incr. | 2437 | 6927 |
| Cal Lin. decr. | 2271 | 10727 | RTL incr. | 4457 | 8838 |
| Cal Lin. conv. decr. | 2304 | 10593 | RTL all | 3543 | 8315 |
| | | | Cal Lin. incr. | 3589 | 8270 |
| | | | Cal Lin. all | 3617 | 8189 |

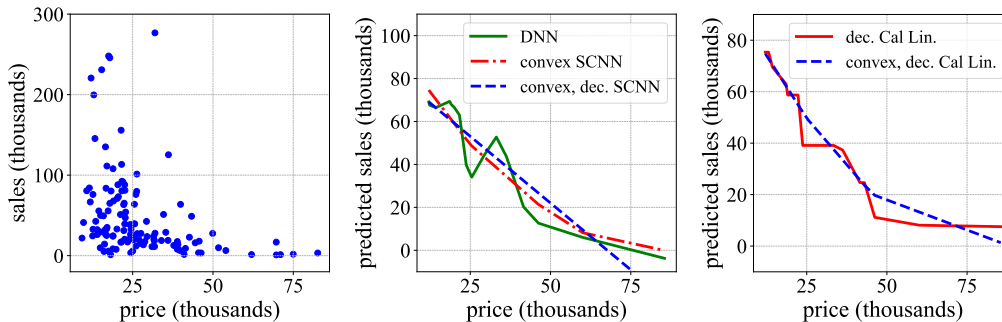

Figure 2: Car sales prediction task. **Left:** Training dataset. **Center:** Predictions for neural nets; y-axis zoomed-in. **Right:** Predictions for calibrated linear models; y-axis zoomed-in.

## 5.2 Puzzle Sales from Reviews

For this small problem (3 features, 156 training, 169 validation, and 200 non-IID test examples, dataset courtesy of Artifact Puzzles and available at `www.kaggle.com/dbahri/puzzles`), we predict the 6-month sales of different wooden jigsaw puzzles from three features based on its Amazon reviews: its average star rating, the number of reviews, and the average word count of its reviews. Business experts expect star rating to have a positive effect on sales, the number of reviews to have a diminishing returns effect on sales, and word count to have a diminishing returns shape (the 100th word is not as important as the 10th word). The train/val/test data is non-IID in that it is split by time over the past 18 months, and the set of puzzles is the same across the datasets, with some new puzzles added over time. Fig. 3 shows the number of reviews calibrator learned for the RTL models. Table 1 shows the Test MSE is slightly better with the added shape constraint for all three models (SCNN, RTL, Cal. Linear). The SCNN concave increasing model is constrained to be concave in both word count and number of reviews and increasing in number of reviews (but does not shape-constrain the star rating feature because we cannot use the SCNN to constrain features positively unless we constrain them to also be concave). The lattice models either imposed just the monotonicity constraints, or all the expected constraints.

## 5.3 Domain Name Pricing

In this experiment (18 features, 1,522 training, 435 validation, and 217 test examples), we illustrate the use of concavity/convexity constraints without any monotonicity constraints. The goal is to learn a model that can automatically price domain names for Google's .app domain. The label is the percent of humans who rated each example domain name as "premium," vs "non-premium." Estimates for new domains were then quantized into pricing tiers. One feature is the number of characters in the domain name, and our experts believe that $f(x)$ should be concave and non-monotonic in this feature. The other features measure the popularity of the ngrams in the domain name according to different internet services.

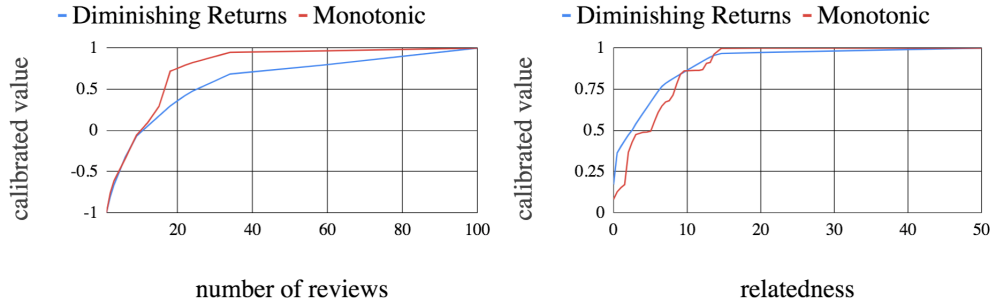

Figure 3: Example calibrator curves learned by the lattice models. The diminishing returns calibrators (blue) are easier to interpret and summarize, and appears less over-fit than the monotonic calibrators (red). **Left:** Calibrated linear model's calibrators for number of reviews for the Puzzles Sales problem. Most of the training examples are in the [1,15] input range, where the curves for the two trained lattice models are very similar. **Right:** RTL model's calibrators for the relatedness feature for the Query Result Matching problem.

Table 2: Experimental Results: Domain Pricing and Wine Quality

| Domain Pricing | | | Wine Quality | | |
|---|---|---|---|---|---|
| Model | Val. MSE | Test MSE | Model | Val. MSE | Test MSE |
| DNN | 0.00301 | 2.00 | DNN | 4.91 | 4.79 |
| SCNN conc. | 0.00220 | 0.00219 | SCNN conc. | 5.96 | 7.22 |
| RTL unc. | 0.1014 | 0.1109 | SCNN conc. incr. | 6.13 | 6.21 |
| RTL conc. | 0.0978 | 0.1078 | RTL incr. | 4.96 | 4.85 |
| Cal Lin. unc. | 0.1101 | 0.1150 | RTL conc. incr. | 4.96 | 4.83 |
| Cal Lin. conc. | 0.1091 | 0.1120 | Cal Lin. incr. | 5.25 | 5.10 |
| | | | Cal Lin. conc. incr. | 5.23 | 5.10 |

The results in Table 2 show the extra concavity constraint slightly improves both the validation error and the test error for the RTL and Calibrated Linear model. The DNN's 2.00 test MSE was an unlucky run; we re-trained the DNN with the same validated hyperparameters 100 times, and only saw test MSE that high 6 times. Similarly, the SCNN got very lucky, when we re-trained it 100 times with the same validated hyperparameters, its test MSE was only as low as shown in Table 2 for 5 of the 100 times. For more data on re-training churn for these models, see Appendix E.

## 5.4 Wine Enthusiast Magazine Reviews

The goal is to predict a wine's quality measured in *points* [80, 100] based on price (the most important feature), country (21 Bools), and 39 Bool features based on the wine description from Wine Enthusiast Magazine (61 features, 84,642 training, 12,092 validation, and 24,185 test examples; www.kaggle.com/dbahri/wine-ratings). Table 2 shows that constraining the price feature does not have much effect on the Test MSE for the RTL and Calibrated Linear models; visual inspection of the learned calibrators (not shown) showed they both picked up the correct shape with or without the shape constraint. The SCNN models had a difficult time fitting this dataset.

## 5.5 Query-Result Matching

The goal of this problem is to learn how well a candidate result matches a query, for a particular category of queries. The dataset (1,282,532 training, 183,219 validation, 366,440 IID test, with 15 features) is proprietary. The 15 features are derived for each {query, result} example, and the label is an averaged human rating of the match quality, from [0, 4].

We give results in Table 3 using the full data for train/validation, and using only 1/10 of the train/validation data; the test set is the same throughout. We would like to constrain 14 of the $D = 15$ features to be monotonic, based on prior knowledge and policies about how the features should impact the output. We constrain the most important feature, *relatedness* to be concave, based on observing that the shape of its calibrator in an unconstrained calibrated linear model appears to exhibit noisy diminishing returns, as shown in Figure 2. For the SCNN with diminishing returns, only the concave feature is constrained to be monotonic.

The biggest effect of the shape constraints is for the calibrated linear models with the smaller training set. The RTL test MSE is hurt a little by the shape constraints on the full training set (recall that even the *unconstrained* RTL model does constrain its calibrators to be monotonic, so all the RTL models get the effect of the monotonic calibrated linear model). We believe this is because there are some parts of the feature space that are very sparse, and satisfying the monotonicity constraints everywhere reduces the RTL's ability to use all its flexibility to do better on average by better fitting the denser parts of the feature space. In practice the test distribution is not IID with the train distribution, and we consider it a worthwhile trade-off to have the model constrained to behave sensibly throughout the potential feature space to protect against embarrassing errors and improve debuggability.

As in our other experiments, the lattice models were more stable across re-trainings than the neural nets: the test MSE standard deviation with the 1/10 train set and each model's validated hyperparameters over five re-trainings was .002 for the RTL models, .084 for the SCNN concave, .047 for the SCNN dim. ret., and 1.69 for the DNN. Surprisingly, the dim. ret SCNN actually does worse than the concave SCNN, even though the constrained feature is definitely a strongly positive signal, but we believe the SCNN MSE differences merely reflect the randomness in training and hyperparameter choices.

Table 3: Experimental Results: Query Result Matching

| Model | 1/10 Train Data | | Full Train Data | |
|---|---|---|---|---|
| | Val. MSE | Test MSE | Val. MSE | Test MSE |
| DNN | 0.668 | 0.668 | 0.655 | 0.656 |
| SCNN conc. | 0.667 | 0.673 | 0.652 | 0.658 |
| SCNN dim. ret. | 0.673 | 0.680 | 0.659 | 0.667 |
| RTL unc. | 0.661 | 0.658 | 0.639 | 0.639 |
| RTL mono | 0.663 | 0.662 | 0.655 | 0.654 |
| RTL all | 0.663 | 0.661 | 0.655 | 0.654 |
| Cal. Lin. unc. | 0.718 | 0.756 | 0.715 | 0.743 |
| Cal. Lin. mono | 0.699 | 0.710 | 0.701 | 0.701 |
| Cal. Lin. all | 0.696 | 0.702 | 0.724 | 0.722 |

## 6   Conclusions

We specified the additional constraints needed to learn flexible lattice models that can impose any mixture of convexity/concavity/monotonicity constraints over subsets of features, ceterus paribus, and showed we can stably train models with these extra linear inequality constraints. The additional shape constraints produce smoother models that are easier to summarize, explain and debug, because their behavior is more predictable and is known to satisfy the specified global properties. Experimental results on real-world problems showed the extra convexity/concaveity shape constraints either reduced or did not affect test MSE on IID test sets, and provided the most value for the non-IID Puzzles experiment, where shape constraint regularization is expected to be most valuable. We also extended neural networks to handle partial diminishing returns constraints. The resulting SCNNs enable a less flexible menu of shape constraint choices, and their experimental results were more sensitive to hyperparameter choices and stochasticity in mini-batch sampling.

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
