[Supplementary Material]

## A    Diminishing Returns for Fine-grained Lattices

We limit our experiments to $2 \times 2 \times 2 \dots$ lattices, which are single-celled lattices. Multi-cell lattices require additional linear inequality constraints on the lattice parameters. Even with multilinear interpolation, to ensure decreasing slopes as the input $x[d]$ moves between the cells of the lattice requires $(M-1) \times 2^{S-1}$ linear inequality constraints for a $M^S$ lattice.

## B    Diminishing Returns for Deep Lattice Networks

To constrain a deep lattice network (DLN) to be convex and increasing for a feature, one must make each layer convex and increasing for any values that are influenced by inputs constrained by those shape constraints. DLN layers are of three types: linear embeddings, 1-d calibrators, or ensembles of lattices. All three layers can be constrained for shape constraints by training with the noted extra linear inequality constraints in this paper. However, if a linear embedding is used before a lattice, for example the $W_l$ matrix in (5), then one must constrain $W_l$ so that each feature only appears in one input to the lattice, because if a feature is part of two inputs to the lattice, then the multilinear interpolation is no longer linear with respect to the feature, which may affect the model's overall convexity or concavity.

## C    Concave/Convex Lattices with Simplex Interpolation

Simplex interpolation (also known as the Lovász extension [29]), is a more efficient way to linearly interpolate a lattice than multilinear interpolation [24], but unlike multilinear interpolation, the produced function is piecewise linear with $S!$ pieces. And as one varies an input $x[d]$, one crosses to different planar pieces, each which have different slopes defined by the lattice parameters. One must constrain the lattice parameters so that all $S!$ planar pieces one crosses have decreasing slope as $x[d]$ goes from $0 \to 1$ to guarantee a concave $f(x)$. The resulting constraints are illustrated in Fig. 4 for $S = 2$ and $S = 3$. In general one must satisfy $(S-1)2^{(S-2)}$ linear inequality constraints to guarantee concavity for a given input.

Figure 4: **Left:** A $S = 2$ dimensional lattice. Simplex interpolation interpolates the $S + 1$ lattice parameters for the upper triangle and bottom triangle, fitting a plane to each of the two simplices. Thus the upper triangle has slope $\theta_4 - \theta_3$, and the lower triangle has slope $\theta_2 - \theta_1$. To guarantee convexity from left-to-right, one must guarantee decreasing slopes by satisfying the linear inequality constraint $\theta_4 - \theta_3 > \theta_2 - \theta_1$. **Right:** For a $S = 3$ dimensional lattice, simplex interpolation is piecewise linear on six simplices, and concavity from left-to-right requires decreasing slopes across adjacent simplices, so four linear inequalities must be satisfied: $\theta_8 - \theta_7 > \theta_4 - \theta_3$, $\theta_8 - \theta_7 > \theta_6 - \theta_5$, $\theta_4 - \theta_3 > \theta_2 - \theta_1$ and $\theta_6 - \theta_5 > \theta_2 - \theta_1$.

Table 4: Simulation

| Model | Validation MSE | Test MSE |
|---|---|---|
| DNN | $1.7 \times 10^{-4}$ | $2.15 \times 10^{-4}$ |
| SCNN concave | $9.9 \times 10^{-4}$ | $1.19 \times 10^{-3}$ |
| SCNN dim. ret. | $5.7 \times 10^{-4}$ | $6.25 \times 10^{-4}$ |
| RTL dim. ret. | $6.0 \times 10^{-4}$ | $5.79 \times 10^{-4}$ |
| Cal. Lin. dim. ret. | $1.98 \times 10^{-1}$ | $1.93 \times 10^{-1}$ |

## D   Simulation

To verify that all methods are working as intended, we learned the following smooth function $f : \mathbb{R}^6 \to \mathbb{R}$ that is ceterus paribus concave and increasing in variables $x[1]$, $x[2]$, $x[3]$:

$$f(x) = \frac{x[1]^{0.3} x[2]^{0.6} x[3]^{0.9}}{e^{x[4]+2x[5]+3x[6]}} + \cos\left(x[4]x[5]x[6]\right) \log\left(\sum_{i=1}^{6} x[i]\right). \tag{6}$$

Examples were sampled on a uniform grid on $[0.5, 1.5]^6$, 5 points per dimension, for a total of $5^6$ points. They were then randomly shuffled and split into training, validation and test sets and used to evaluate adding concavity and diminishing returns constraints on $x[1], x[2], x[3]$ for various function classes. Results in Table 4 verify that all methods are working as expected, and that the methods with the additional shape constraints show less overfitting as measured by the difference in validation and test MSE.

## E   Churn Data for Domain Name Pricing Results

We ran follow-up experiments to better characterize the re-training churn [30] of the test results for the DNN, SCNN, and RTL models for the Domain Pricing experiment with respect to re-training. We fixed the hyperparameters to be the ones chosen by the validation process. Then we re-trained each model type 100 times. We sorted the resulting 100 test MSE's for each model type, and then plotted the sorted test MSE's in Figure 5.

The 100 re-trainings differ because for both model types, each re-training experienced a different random shuffle of the training data, and randomized mini-batching of the stochastic gradients. For DNN, there is also randomness from the initialization: the models were initialized using Glorot initialization, that is, the weights are uniform random over some interval, and the biases are initialized to 0. For SCNN, the models were initialized deterministically using the identity matrix and zero for the biases. The RTL models are deterministically initialized based on the training data quantiles, this leads to very stable results across re-trainings.

The DNN results show the most re-training variability. Over the 100 re-trainings, the DNN test MSE ranges from 0.00143 to 2.002, with a mean of 0.27, and median 0.05. For the SCNN, the deterministic regularization and added regularization from the shape constraints does seem to upper-bound the test error, but it still has a large range of .00122 to 0.3161, with better mean of 0.072 and worse median of 0.069.

## F   TensorFlow Code for Shape-Constrained Neural Network

```
1  import tensorflow as tf
2
3  def scnn_model_fn(features, labels, mode, params):
4    inputs = {}
5    for input_name in ['xu', 'xc', 'xs']:
6      if input_name not in features:
7        continue
8      input_dim = features[input_name].shape[1].value
```

Figure 5: Plots show the 100 sorted test MSE values for 100 different DNN, SCNN and RTL models, where each of the models was re-trained with the (same) hyperparameters chosen on the validation set.

```
9      if input_dim > 0:
10       inputs[input_name] = tf.feature_column.input_layer(
11         {input_name: features[input_name]},
12         [tf.feature_column.numeric_column(
13           input_name, shape=(input_dim)
14         )])
15
16    # for simplicity, assume at least one feature is unconstrained
17    assert 'xu' in inputs
18    fc = tf.layers.dense
19
20    # if 'is_convex', constrain xc and xs to be convex, else concave
21    g = (tf.nn.relu if params['is_convex']
22         else lambda x: -1*tf.nn.relu(-1*x))
23    h = tf.nn.relu
24
25    u_dims = params['u_dims']
26    z_dims = params['z_dims']
27    n_layers = len(z_dims)
28    nonneg = lambda x: tf.maximum(x, 0.0)
29    bias_init = tf.constant_initializer(0.0)
30    kernel_init = tf.initializers.identity()
31
32    prev_u = inputs['xu']
33    for i in range(n_layers):
34      z_dim = z_dims[i]
35
36      pre = fc(prev_u, z_dim, kernel_initializer=kernel_init,
37               bias_initializer=bias_init)
38
39      if 'xs' in inputs:
40        inner = fc(prev_u, inputs['xs'].shape[1],
41                 kernel_initializer=kernel_init,
42                 bias_initializer=bias_init,
43                 activation=tf.nn.relu)
```

```
44        pre += fc(tf.multiply(inputs['xs'], inner),
45                  z_dim, kernel_constraint=nonneg,
46                  kernel_initializer=kernel_init,
47                  use_bias=False)
48
49      if 'xc' in inputs:
50        inner = fc(prev_u, inputs['xc'].shape[1],
51                   kernel_initializer=kernel_init,
52                   bias_initializer=bias_init)
53        pre += fc(tf.multiply(inputs['xc'], inner),
54                  z_dim, kernel_initializer=kernel_init,
55                  use_bias=False)
56
57      if i > 0:
58        prev_z_dim = z_dims[i-1]
59        inner = fc(prev_u, prev_z_dim, kernel_initializer=kernel_init,
60                   bias_initializer=bias_init,
61                   activation=tf.nn.relu)
62        pre += fc(tf.multiply(prev_z, inner), z_dim,
63                  kernel_constraint=nonneg,
64                  kernel_initializer=kernel_init,
65                  use_bias=False)
66
67      if i == n_layers-1:
68        z = pre
69      else:
70        z = g(pre)
71
72      prev_z = z
73      if i != n_layers-1:
74        prev_u = fc(prev_u, u_dims[i],
75                    kernel_initializer=kernel_init,
76                    bias_initializer=bias_init,
77                    activation=h)
78
79      if mode == tf.estimator.ModeKeys.TRAIN:
80        optimizer = tf.train.AdamOptimizer(
81          learning_rate=params['learning_rate'])
82        loss = tf.losses.mean_squared_error(z, labels)
83        train_op = optimizer.minimize(
84          loss, global_step=tf.train.get_global_step())
85        return tf.estimator.EstimatorSpec(
86          mode, loss=loss, train_op=train_op)
87      if mode == tf.estimator.ModeKeys.PREDICT:
88        predictions = {'predictions': z}
89        return tf.estimator.EstimatorSpec(mode, predictions=predictions)
90
91 # example usage
92 scnn_estimator = tf.estimator.Estimator(
93   model_fn=scnn_model_fn, params={'u_dims': [50], 'z_dims':[50, 1],
94   'learning_rate': 0.1, 'is_convex': False})
```