[Reviews · NeurIPS 2018]

Reviewer 1



This paper adds diminishing / accelerating return constraints to lattice models. The authors showed with the additional knowledge of features, their approach can reduce the testing error on some datasets. I think the idea of the paper, that to utilize human knowledge when the data is scarce, is very interesting. Utilizing diminishing / accelerating return constraints is novel according to my knowledge.

Reviewer 2



The authors argue that accelerating/decelerating return constraints can help model interpretability. To this end, they propose two methods: (i) an extension of input convex neural networks that supports monotonicity constraints; (ii) an extension of partial monotonic lattice models that supports concavity/convexity constraints. The proposed methods are compared to several constrained and unconstrained baselines on different datasets. Pros: - The paper is very well written, and definitely relevant to NIPS - Constraining the model to have accelerating returns w.r.t. chosen variables allows to better control the model, which can be useful in real applications - The technical contribution seems sound - The empirical evaluation is reasonably thorough (several datasets and relevant competitors) Cons: - The term "interpretability" is unqualified, little evidence for it is provided - The improvement in terms of pure performance is minor or negligible - All datasets have a small number of features; scalability and stability of learning are not evaluated Major remarks: - The only points that I am really concerned about are the first two. It seems that the term "interpretability" here is used as a synonym of "visualizability". I suggest the authors to clarify this in the text. It also seems to me that visualizability only applies to the calibrator functions. Indeed, for higher-dimensional models the interactions between variables may be very complicated, making it difficult or impossible to show the effect of inputs on the output. In other words, I think that visualizability does not really apply to the constrained SCNN and RTL models as a whole. This seems to mine the idea that these constraints can improve visualizability (at least in higher dimensional models). It is still the case that the constraints help in controlling the model, and this is good; but this is *different* from interpretability. As a matter of fact, if the learned functions can not be visualized, the interpretability argument becomes moot: the user already knows that the curves will obey the constraints that she specified -- but she won't be able to see what the curves actually look like, i.e., she won't be able to really analyze and understand them. The authors should qualify their claims, evaluate them empirically (e.g. with a user study), or drop them. - I agree that MSE is not the only useful property of the model. In some applications it makes sense to sacrifice performance in exchange for more control. However, in the four datasets (with the caveat that MSE is non-trivial to interpret...): - Cal Sales is essentially a toy dataset (it's 1D), it's useful for debugging but not really representative of most ML applications. - Puzzle Sales (3 features): all models do "much worse" than the DNN baseline; constraints have some positive impact, but nothing major. Note that the test set is non-IID, so these numbers may be biased. - Domain Pricing (18 features): performance is better than the baseline (which however overfits horribly), but the additional constraints have little impact. - Wine Quality (61 features): performance of SCNN improves slightly, lattice performance is unaffected. - Ranking (15 features): considering only MSE on the IID test set [the non-IID numbers are much harder to analyze], SCNN and RTL performance worsens slightly for the full train data case. It seems that the more features there are, the less evidence there is for the additional constraints to really help performance. In a way, the paper is missing a "killer application" where shape constraints can really make performance better. However, as I said, the contribution makes sense anyway, as it allows to better control the learned model. I think that the authors should clarify that (ultimately) the contribution is not really about performance. Minor remarks: - The observation in Appendix D seems quite important to me (it is a limitation of the model, after all), and should be moved to the main text for visibility. - What is the impact on run-time of adding the constraints? What is the impact on learning stability? - (It would be nice to check what is the effect of these constraints on adversarial examples.)

Reviewer 3



This paper is about learning models constrained to be convex/concave increasing/decreasing in their *inputs*. Two such models are studied: neural networks, and lattice interpolation models. The two models are not comparable: the former is *jointly* constrained, while the latter is independently constrained in each input dimension. The core motivation is interpretability and encoding prior assumptions (e.g. diminishing returns.) The learned models strictly enforce their constraints and perform well (sometimes better than unconstrained models) on a number of real-world tasks (albeit fairly low-dimensional -- a limitation of lattice models, if I understand correctly). Strengths: - Two new constrained architectures with different properties but good performance are proposed. This gives potential users choice. - The exposition and motivatoin of the paper are very good: a pleasant and convincing read for someone who has not previously thought much about enforcing such constraints. It will change the way I tackle certain problems in the future. Weaknesses: - While there is not much related work, I am wondering whether more experimental comparisons would be appropriate, e.g. with min-max networks, or Dugas et al., at least on some dataset where such models can express the desired constraints. - The technical delta from monotonic models (existing) to monotonic and convex/concave seems rather small, but sufficient and valuable, in my opinion. - The explanation of lattice models (S4) is fairly opaque for readers unfamiliar with such models. - The SCNN architecture is pretty much given as-is and is pretty terse; I would appreciate a bit more explanation, comparison to ICNN, and maybe a figure. It is not obvious for me to see that it leads to a convex and monotonic model, so it would be great if the paper would guide the reader a bit more there. Questions: - Lattice models expect the input to be scaled in [0, 1]. If this is done at training time using the min/max from the training set, then some test set samples might be clipped, right? Are the constraints affected in such situations? Does convexity hold? - I know the author's motivation (unlike ICNN) is not to learn easy-to-minimize functions; but would convex lattice models be easy to minimize? - Why is this paper categorized under Fairness/Accountability/Transparency, am I missing something? - The SCNN getting "lucky" on domain pricing is suspicious given your hyperparameter tuning. Are the chosen hyperparameters ever at the end of the searched range? The distance to the next best model is suspiciously large there. Presentation suggestions: - The introduction claims that "these shape constraints do not require tuning a free parameter". While technically true, the *choice* of employing a convex or concave constraint, and an increasing/decreasing constraint, can be seen as a hyperparameter that needs to be chosen or tuned. - "We have found it easier to be confident about applying ceterus paribus convexity;" -- the word "confident" threw me off a little here, as I was not sure if this is about model confidence or human interpretability. I suspect the latter, but some slight rephrasing would be great. - Unless I missed something, unconstrained neural nets are still often the best model on half of the tasks. After thinking about it, this is not surprising. It would be nice to guide the readers toward acknowledging this. - Notation: the x[d] notation is used in eqn 1 before being defined on line 133. - line 176: "corresponds" should be "corresponding" (or alternatively, replace "GAMs, with the" -> "GAMs; the") - line 216: "was not separately run" -> "it was not separately run" - line 217: "a human can summarize the machine learned as": not sure what this means, possibly "a human can summarize what the machine (has) learned as"? or "a human can summarize the machine-learned model as"? Consider rephrasing. - line 274, 279: write out "standard deviation" instead of "std dev" - line 281: write out "diminishing returns" - "Result Scoring" strikes me as a bit too vague for a section heading, it could be perceived to be about your experiment result. Is there a more specific name for this task, maybe "query relevance scoring" or something? === I have read your feedback. Thank you for addressing my observations; moving appendix D to the main seems like a good idea. I am not changing my score.